# Multi-Agent Reasoning with Consistency Verification Improves Uncertainty Calibration in Medical MCQA

## Abstract

Miscalibrated confidence scores are a practical obstacle to deploying AI in clinical settings. A model that is always overconfident offers no useful signal for deferral. We present a multi-agent framework that combines domain-specific specialist agents with Two-Phase Verification (Wu et al., 2024) and S-Score Weighted Fusion to improve both calibration and discrimination in medical multiple-choice question answering. Four specialist agents (respiratory, cardiology, neurology, gastroenterology) generate independent diagnoses using Qwen2.5-7B-Instruct. Each diagnosis is then subjected to a two-phase self-verification process that measures internal consistency and produces a Specialist Confidence Score (S-score). The S-scores drive a weighted fusion strategy that selects the final answer and calibrates the reported confidence. We evaluate on high-disagreement subsets of MedQA-USMLE and MedMCQA, covering 100-question and 250-question settings. All reported results are specific to this filtered regime and should not be extrapolated to the full benchmark distributions. On MedQA-250, the primary evaluation, the full system achieves ECE = 0.091 (74.4% reduction over the single-specialist baseline) and AUROC = 0.630 (+0.056) at 59.2% accuracy. Calibration gains of 49–74% hold across all four settings, including MedMCQA where they persist even when absolute accuracy is constrained by knowledge-intensive recall demands. Ablation analysis reveals that calibration and discrimination respond to distinct components of the framework. Two-Phase Verification is the primary driver of ECE reduction while multi-agent reasoning drives AUROC improvement. This decomposition suggests that inference-time consistency checking and ensemble aggregation address different failure modes of LLM uncertainty, a finding that may generalize beyond the medical QA setting studied here. Whether the resulting confidence signal is sufficient to support clinical deferral decisions in practice remains a direction for future investigation rather than a demonstrated capability.

## 1 Introduction

Confidence scores from medical AI systems carry real consequences. An overconfident wrong answer may go unchallenged, while a system that assigns uniformly low confidence to every prediction becomes clinically useless. Yet current large language models (LLMs) are systematically miscalibrated, producing confidence scores that poorly track actual correctness (Guo et al., 2017; Kadavath et al., 2022). On specialised, knowledge-intensive medical benchmarks this miscalibration is especially pronounced (Pal et al., 2022; Jin et al., 2021).

Existing recalibration methods range from post-hoc temperature scaling (Guo et al., 2017) and deep ensembles (Lakshminarayanan et al., 2017) to Bayesian dropout (Gal & Ghahramani, 2016), but each requires either labeled calibration data or multiple model copies. Wu et al. (2024) proposed a label-free alternative in which Two-Phase Verification decomposes a model's reasoning into factual claims, answers them independently and then with reference, and uses cross-condition inconsistency as an uncertainty signal. Because it operates purely at inference time on the model's own outputs, it is well-suited to deployment settings where labeled data are scarce.

Separately, multi-agent LLM systems have shown accuracy gains by pooling diverse specialist perspectives (Wang et al., 2024b; Li et al., 2023), yet how aggregation affects confidence calibration has not been studied. Our work brings these two threads together. We propose **MARC** (**M**ulti-**A**gent **R**easoning with **C**onsistency Verification), a framework with three components. Domain-specific specialist agents provide diverse perspectives on each question. Two-Phase Consistency Verification (Wu et al., 2024) converts per-specialist internal consistency into a Specialist Confidence Score (S-score). S-Score Weighted Fusion then aggregates specialist votes weighted by verified confidence to select the final answer and calibrate the reported confidence.

The primary contribution is **MARC** itself, a multi-agent architecture that improves both calibration (ECE) and discrimination (AUROC) over single-agent baselines by using consistency-derived S-scores as fusion weights, replacing heuristic aggregation with an uncertainty-aware signal. A four-configuration ablation across four experimental settings (MedQA-100, MedQA-250, MedMCQA-100, MedMCQA-250) isolates Two-Phase Verification as the primary calibration driver and multi-agent reasoning as the primary accuracy driver. The study also yields an empirical characterization of the knowledge-recall versus clinical-reasoning divide between MedMCQA and MedQA, with implications for how far verification-based calibration can reach in small LLMs. A central empirical observation is that ECE and AUROC respond to fundamentally different components of the framework. Verification compresses overconfident scores toward the model's actual accuracy level, while multi-agent fusion sharpens the rank ordering that separates correct from incorrect predictions. This decomposition is the general finding that the medical QA setting allows us to characterize, and it may transfer to other domains where inference-time uncertainty quantification without labeled calibration data is needed.

## 2 Related Work

### 2.1 Uncertainty Quantification in Medical AI

Uncertainty quantification (UQ) is essential in clinical AI for safe deployment and appropriate deferral to human experts (Kompa et al., 2021; Begoli et al., 2019). Ensemble methods (Lakshminarayanan et al., 2017) and Monte Carlo dropout (Gal & Ghahramani, 2016) provide well-studied UQ baselines but require multiple forward passes or model modifications. Temperature scaling (Guo et al., 2017) is a simple post-hoc recalibration technique that rescales model logits, achieving good calibration at low cost but without improving discrimination. Conformal prediction (Angelopoulos & Bates, 2021) offers distribution-free coverage guarantees but focuses on set-valued predictions rather than scalar confidence scores.

### 2.2 LLM Self-Evaluation and Verification

LLMs can assess their own outputs through self-consistency sampling (Wang et al., 2023), chain-of-thought verification (Wei et al., 2022b), and explicit self-critique (Madaan et al., 2023). Xiong et al. (2024) found that LLMs tend to be overconfident when verbalizing uncertainty, and that no single elicitation strategy consistently outperforms others across tasks requiring professional knowledge. Tian et al. (2023) showed that for RLHF fine-tuned models such as ChatGPT and GPT-4, verbalized confidence scores are better calibrated than token-level log-probabilities, reducing ECE by up to 50% on open-domain QA benchmarks. Wu et al. (2024) introduced Two-Phase Verification specifically for medical QA, in which reasoning is decomposed into factual claims that are answered independently and with reference, and the cross-condition inconsistency measures epistemic uncertainty without requiring gold labels. We adopt this method as our per-specialist UQ mechanism.

### 2.3 Multi-Agent Systems for Medical QA

Multi-agent LLM systems have demonstrated accuracy improvements on complex reasoning tasks by combining diverse agent perspectives (Wang et al., 2024b; Li et al., 2023; Liang et al., 2024). In the medical domain, Wang et al. (2024a) showed that an LLM-based multi-specialist consultation framework improves automatic diagnosis accuracy over single-agent baselines. Du et al. (2024) demonstrated that multiple agents debating their answers can reduce reasoning errors. However, these works focus almost exclusively on accuracy, and the calibration properties of multi-agent aggregation have not been systematically studied.

### 2.4 Medical QA Benchmarks

MedQA-USMLE (Jin et al., 2021) contains about 12,700 US Medical Licensing Examination-style questions with clinical vignettes averaging 109 words, testing integrated clinical reasoning across 4 options. MedMCQA (Pal et al., 2022) contains over 194,000 questions drawn from two Indian postgraduate medical entrance examinations, AIIMS PG and NEET-PG, with short stems averaging 12.8 tokens across 21 specialties. GPT-4 clears the USMLE passing threshold by a wide margin (Nori et al., 2023) and recent frontier models exceed 90% on MedQA, but scores are markedly lower on MedMCQA, where specialised factual recall dominates (Pal et al., 2022; Singh et al., 2025). Recent work highlights that smaller 7B-class models exhibit pronounced accuracy drops on knowledge-intensive recall questions, independent of reasoning capability (Singh et al., 2025; Wei et al., 2022a).

## 3 Methodology

### 3.1 System Architecture

Given a medical multiple-choice question with options, four specialist agents generate independent diagnoses. Each diagnosis undergoes Two-Phase Verification to produce an S-score. Finally, S-Score Weighted Fusion selects the answer and reports a calibrated confidence. Figure 1 illustrates the full MARC pipeline.

### 3.2 Specialist Agents

We instantiate four domain-specific specialist agents — respiratory (pulmonologist), cardiology (cardiologist), neurology (neurologist), and gastroenterology (gastroenterologist) — covering the specialties that appear most frequently in the high-disagreement subsets of both benchmarks.

Each agent is driven by a Qwen2.5-7B-Instruct model (Qwen Team, 2024) with a specialty-specific system prompt that instructs the agent to reason from their domain expertise using a five-step chain-of-thought (CoT) protocol: (1) clinical scenario analysis, (2) differential diagnosis, (3) systematic option evaluation, (4) option comparison, and (5) final decision with confidence score. Greedy decoding ($T = 0$) is used for all specialist calls, while verification sub-calls use temperature sampling with deterministic seeds (see Section 3.5).

### 3.3 Two-Phase Verification

Following Wu et al. (2024), each specialist first generates answer $a_k$ with step-by-step explanation $r_k$ and initial confidence $C_k^{(0)} \in [0, 1]$ (Phase 1). Phase 2 then proceeds as follows:

1. *Question formulation.* Four verification questions $\{q_j\}_{j=1}^4$ are extracted from $r_k$ by prompting the model to identify key factual claims.

2. *Independent answering.* Each $q_j$ is answered without reference to $r_k$, yielding answers $\{a_j^{\text{ind}}\}$.

3. *Reference answering.* Each $q_j$ is answered with $r_k$ in context, yielding $\{a_j^{\text{ref}}\}$.

4. *Inconsistency measurement.* Each pair $(a_j^{\text{ind}}, a_j^{\text{ref}})$ is compared using Jaccard similarity on stop-word-filtered tokens. A pair is considered consistent if the Jaccard score exceeds $\tau = 0.4$, or if the Jaccard score of content words (length $> 4$) exceeds 0.6 (a secondary check that handles cases where one answer is more verbose but carries the same key terms). The inconsistency score over the $N_k \leq 4$ successfully parsed question pairs is:

$$I_k = \frac{1}{N_k} \sum_{j=1}^{N_k} \mathbf{1}\big[\text{sim}(a_j^{\text{ind}}, a_j^{\text{ref}}) < \tau\big] \tag{1}$$

where $\mathbf{1}[\cdot]$ is the indicator function (1 if the condition holds, 0 otherwise), and $N_k = 4$ in the typical case when all questions are successfully parsed.

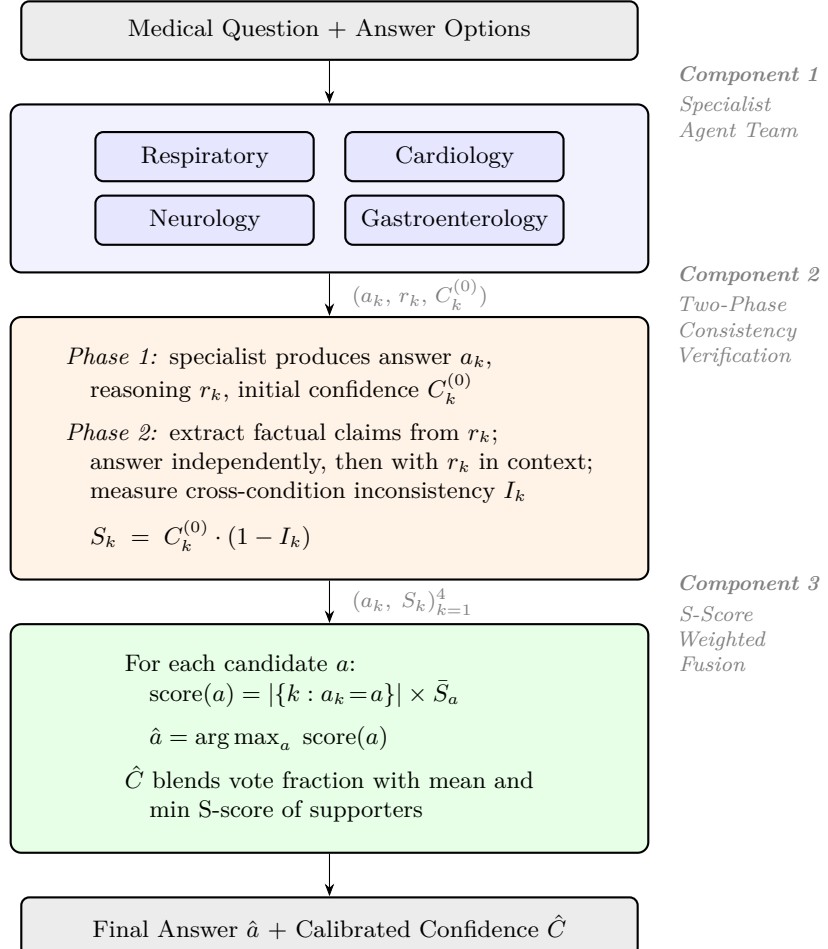

Figure 1: The MARC framework pipeline. A medical question is distributed to four domain-specific specialist agents (Component 1), each producing an answer and reasoning chain. Two-Phase Consistency Verification (Component 2; Wu et al. 2024) extracts factual claims, answers them independently and with the original reasoning in context, and measures cross-condition inconsistency to derive an S-score $S_k$ per specialist. S-Score Weighted Fusion (Component 3) selects the final answer by vote-count-weighted mean S-score and derives calibrated confidence $\hat{C}$ from the vote fraction and S-score distribution.

The Specialist Confidence Score (S-score) is then computed as:

$$S_k = C_k^{(0)} \cdot (1 - I_k) \tag{2}$$

This *multiplicative* formula penalizes initial confidence proportionally to measured inconsistency, so a perfectly consistent specialist ($I_k = 0$) retains its full confidence while a fully inconsistent one ($I_k = 1$) receives $S_k = 0$.

Two alternative formulations were considered and rejected in preliminary experiments. A weighted-average variant $S_k = \alpha C_k^{(0)} + (1 - \alpha)(1 - I_k)$ with $\alpha = 0.65$ decouples the two signals additively but introduces a mixing parameter that requires a labeled validation set to tune. A pure-consistency variant $S_k = 1 - I_k$ discards initial confidence entirely, which degraded performance on questions where the specialist's initial confidence was already well-calibrated. The multiplicative formula was selected because it preserves the sign relationship between initial confidence and the penalty. A high-confidence specialist incurs a larger absolute reduction than a low-confidence one facing the same level of inconsistency, which ensures that well-calibrated specialists are rewarded and overconfident inconsistent ones are penalized most severely.

### 3.4 S-Score Weighted Fusion

Let $\mathcal{K}$ be the set of specialist agents, each proposing answer $a_k \in \{A, B, C, D\}$ with S-score $S_k$. For each candidate answer $a$, we compute an aggregate score:

$$\text{score}(a) = |\{k : a_k = a\}| \times \bar{S}_a \tag{3}$$

where $\bar{S}_a = \frac{1}{|\{k:a_k=a\}|} \sum_{k:a_k=a} S_k$ is the mean S-score of specialists supporting answer $a$, and $|\{k : a_k = a\}|$ is their vote count. The winning answer is $\hat{a} = \arg\max_a \text{score}(a)$.

Final confidence is calibrated by the degree of consensus. When all specialists agree ($|\{a_k\}| = 1$), we use $\hat{C} = \max_k S_k$. Otherwise:

$$\hat{C} = v \cdot \bar{S}_{\hat{a}} + (1 - v) \cdot \min_{k:\, a_k = \hat{a}} S_k \tag{4}$$

where $v = |\{k : a_k = \hat{a}\}|/|\mathcal{K}|$ is the vote fraction and $\bar{S}_{\hat{a}}$ is the mean S-score of winning supporters. This blends mean and minimum S-score weighted by majority strength, yielding lower confidence for weak majorities and higher confidence for strong consensus.

### 3.5 Experimental Setup

**Datasets.** We construct four evaluation sets, all filtered to *high-disagreement* questions, specifically those where our four specialists initially disagree, which maximizes the difficulty and the importance of uncertainty quantification. *MedQA-100* and *MedQA-250* contain 100 and 250 questions sampled from the high-disagreement subset of MedQA-USMLE (Jin et al., 2021), with clinical vignettes averaging 109 words and four answer options. *MedMCQA-100* and *MedMCQA-250* contain 100 and 250 questions from the high-disagreement subset of MedMCQA (Pal et al., 2022), spanning 21 specialties with short stems averaging 15 words. The same specialist-curation and disagreement-labeling pipeline is applied to both benchmarks (see Appendix A.1). These high-disagreement subsets represent the regime where uncertainty quantification is most consequential. Results reported throughout this paper apply specifically to this filtered setting and may not generalize to questions where the specialists are more likely to reach unanimous agreement.

**Model.** All experiments use Qwen2.5-7B-Instruct (Qwen Team, 2024) in full FP16 precision (no quantization), loaded from a local checkpoint for reproducibility, and run on a single NVIDIA GeForce RTX 5090 Laptop GPU (24 GB VRAM). Greedy decoding ($T = 0$) is used for all specialist calls. Temperature sampling with deterministic seeds is used for verification sub-calls ($T_{\text{questions}} = 0.3$, $T_{\text{independent}} = 0.4$, $T_{\text{reference}} = 0.2$).

**Ablation configurations.** Four configurations are compared in a factorial ablation design. *Config 1* is the single-specialist baseline, using the respiratory agent alone with no verification. *Config 2* adds Two-Phase Verification to that same agent, isolating the calibration effect of verification without any multi-agent contribution. *Config 3* uses all four specialists fused with S-Score Weighted Fusion but omits Two-Phase Verification (S-score $= C^{(0)}$), isolating the accuracy contribution of multi-agent reasoning. *Config 4* is the full system, combining four specialists with Two-Phase Verification and S-Score Weighted Fusion. The comparisons $1 \rightarrow 2$, $1 \rightarrow 3$, and $3 \rightarrow 4$ isolate verification, multi-agent reasoning, and the interaction between the two, respectively.

Two aspects of the baseline design deserve explicit attention. When Two-Phase Verification is omitted, each specialist's S-score reduces to its raw initial confidence $C^{(0)}$, and the fusion formula (Eq. 3) becomes confidence-weighted majority voting. Config 3 therefore functions as the majority-vote baseline within the ablation, even though it is not labeled as such in Table 1. Standard post-hoc calibration methods such as temperature scaling (Guo et al., 2017) and isotonic regression require a labeled calibration set to fit their parameters. Because this work targets the inference-only deployment setting where such labels are unavailable, these methods are not applicable baselines and are excluded from comparison.

**Evaluation metrics.** Three metrics are reported. *Accuracy* is the fraction of correctly answered questions. *ECE* (Expected Calibration Error, Guo et al. 2017) measures how well predicted confidence tracks observed accuracy across 5 equal-width bins of width 0.2, with lower values indicating better calibration. Five bins were chosen deliberately. At the smallest evaluation scale of $n = 100$ questions, five bins of width 0.2 each

contain on average 20 observations, providing stable per-bin estimates. Because this choice is held constant across all four configurations and all four datasets, the relative ECE improvements reported in Table 1 are not sensitive to the bin-count selection. *AUROC* is the area under the ROC curve using confidence as the ranking score to separate correct from incorrect answers, with higher values indicating better discrimination.

# 4 Results

## 4.1 Main Results

Table 1 reports performance across all four datasets and four configurations. Figure 2 visualises accuracy, ECE, and AUROC side-by-side for each dataset. The 250-question results (MedQA-250 and MedMCQA-250) constitute the primary evidence, as the larger sample size yields more stable metric estimates. The 100-question results are included as consistency checks at smaller scale and should be interpreted with that caveat in mind. ECE estimates on the 100-question sets carry greater sampling variability, and the calibration improvement on MedMCQA-100 should be read as directional rather than statistically conclusive. Bootstrap 95% confidence intervals for all metrics are reported in Appendix A.3 (Table 3).

Table 1: Performance comparison across all four datasets and configurations. Best value in each column group is **bold**. Δ columns show change relative to Config 1 baseline. The italic *Avg. 4 Specialists* row reports the mean performance across all four individual specialists running independently (see Table 4 for per-specialist details). Avg Conf = mean predicted confidence score. Time = wall-clock inference time in minutes on a single NVIDIA RTX 5090 GPU.

| Dataset | Configuration | Acc | ΔAcc | ECE | ΔECE | AUROC | ΔAUROC | Avg Conf | Time (min) |
|---|---|---|---|---|---|---|---|---|---|
| MedQA-100 | C1: Single Specialist | 52.0% | — | 0.374 | — | 0.537 | — | 0.886 | 39 |
| | *Avg. 4 Specialists* | *54.2%* | — | *0.360* | — | *0.552* | — | — | — |
| | C2: Single + Two-Phase | 52.0% | +0.0 | 0.185 | −51% | **0.678** | +0.141 | 0.430 | 67 |
| | C3: Multi + S-Score (No 2P) | 55.0% | +3.0pp | 0.356 | −5% | 0.578 | +0.041 | 0.896 | 152 |
| | C4: Full System | **59.0%** | +7.0pp | **0.098** | −73% | 0.645 | +0.108 | 0.549 | 249 |
| MedQA-250 | C1: Single Specialist | 54.4% | — | 0.355 | — | 0.574 | — | 0.899 | 116 |
| | *Avg. 4 Specialists* | *57.2%* | — | *0.325* | — | *0.555* | — | — | — |
| | C2: Single + Two-Phase | 54.4% | +0.0 | 0.178 | −50% | 0.556 | −0.018 | 0.424 | 207 |
| | C3: Multi + S-Score (No 2P) | 57.2% | +2.8pp | 0.336 | −5% | 0.587 | +0.013 | 0.905 | 464 |
| | C4: Full System | **59.2%** | +4.8pp | **0.091** | −74% | **0.630** | +0.056 | 0.569 | 833 |
| MedMCQA-100 | C1: Single Specialist | 45.0% | — | 0.469 | — | 0.501 | — | 0.907 | 40 |
| | *Avg. 4 Specialists* | *47.8%* | — | *0.426* | — | *0.540* | — | — | — |
| | C2: Single + Two-Phase | 45.0% | +0.0 | 0.323 | −31% | 0.437 | −0.064 | 0.440 | 65 |
| | C3: Multi + S-Score (No 2P) | **52.0%** | +7.0pp | 0.392 | −16% | **0.541** | +0.040 | 0.902 | 181 |
| | C4: Full System | 50.0% | +5.0pp | **0.237** | −49% | 0.518 | +0.017 | 0.550 | 288 |
| MedMCQA-250 | C1: Single Specialist | 42.8% | — | 0.469 | — | 0.536 | — | 0.897 | 102 |
| | *Avg. 4 Specialists* | *44.7%* | — | *0.446* | — | *0.553* | — | — | — |
| | C2: Single + Two-Phase | 42.8% | +0.0 | 0.271 | −42% | 0.503 | −0.033 | 0.460 | 167 |
| | C3: Multi + S-Score (No 2P) | **46.8%** | +4.0pp | 0.433 | −8% | 0.562 | +0.026 | 0.900 | 409 |
| | C4: Full System | 44.0% | +1.2pp | **0.176** | −63% | **0.594** | +0.058 | 0.548 | 712 |

## 4.2 Calibration Analysis

Two-Phase Verification is the dominant calibration driver. Comparing Config 1 to Config 2 (verification only, same single agent), ECE drops by 51% on MedQA-100, 50% on MedQA-250, 31% on MedMCQA-100, and 42% on MedMCQA-250, with no change in accuracy. Adding verification to the multi-agent system (Config 3→4) yields even larger gains, with 73% (MedQA-100), 73% (MedQA-250), 40% (MedMCQA-100), and 59% (MedMCQA-250) ECE reduction. These improvements are visualised in Figure 3.

Figure 4 provides calibration histograms comparing Config 1 (single specialist baseline) and Config 4 (full system) across all four datasets.

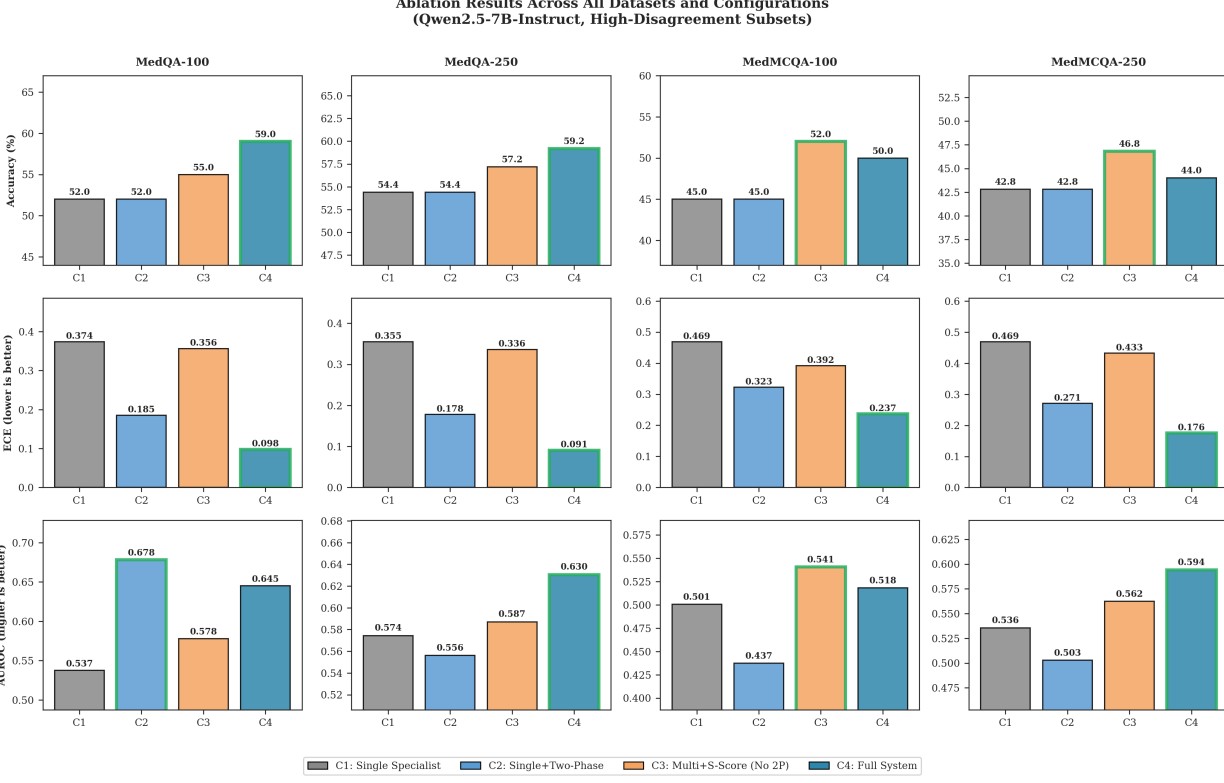

Figure 2: Accuracy (top row), ECE (middle row), and AUROC (bottom row) for each of the four datasets and all four configurations. Green-outlined bars indicate the best value in each panel. Config 4 (Full System, dark blue) achieves the best ECE across all four datasets, and the best accuracy on the MedQA subsets. On MedMCQA, Config 3 achieves the best accuracy at the cost of calibration, illustrating the knowledge-recall challenge for verification-based methods.

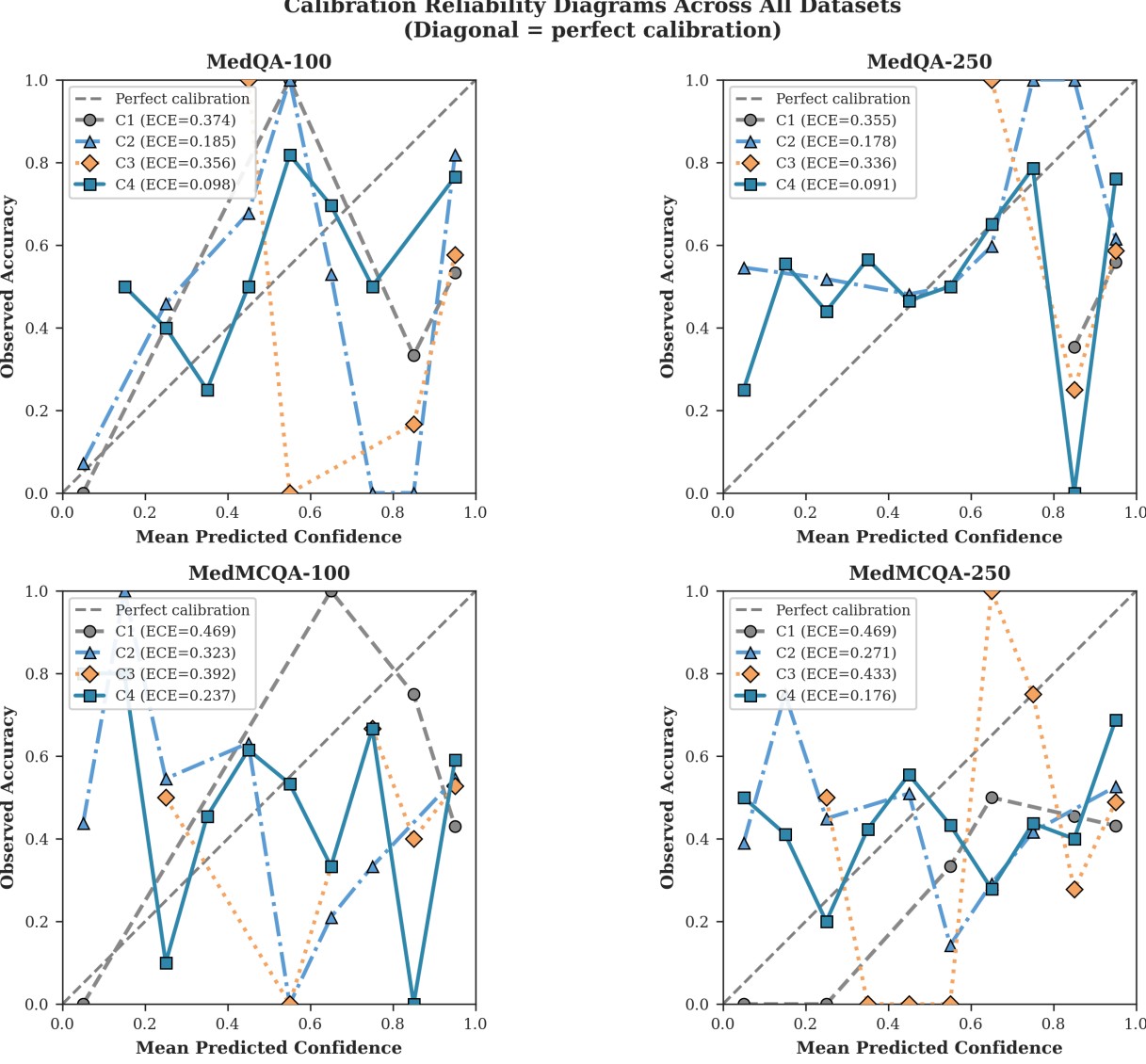

Figure 3: Reliability diagrams for all four datasets. Each panel shows calibration curves for all four configurations against the perfect-calibration diagonal. Across all datasets, Config 4 (solid blue squares) lies closest to the diagonal, confirming that Two-Phase Verification is the dominant calibration driver regardless of question type or dataset size.

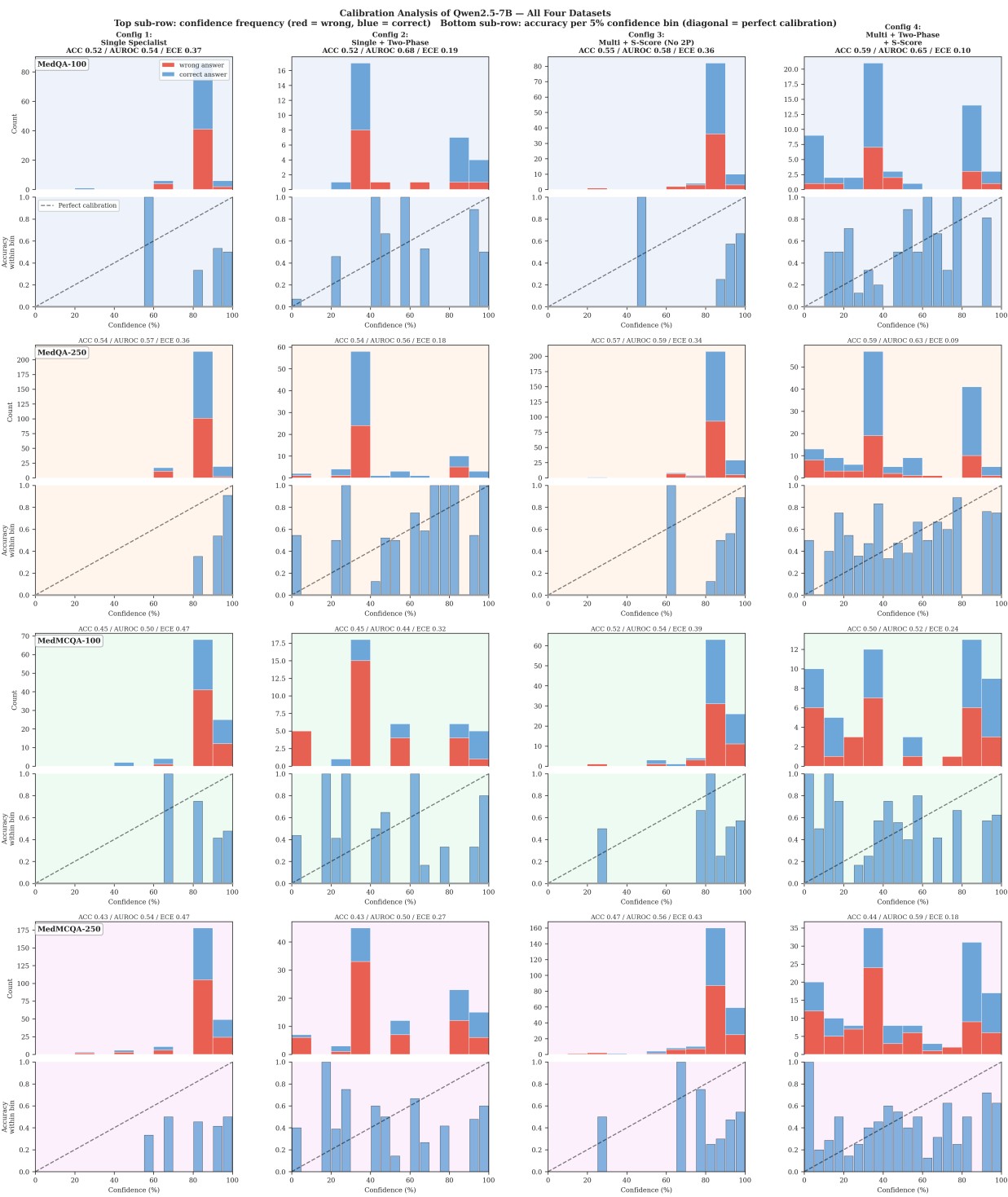

Figure 4: Calibration analysis of Qwen2.5-7B across all four evaluation sets. Columns represent configurations C1–C4; each dataset occupies two sub-rows. *Top sub-row*: stacked confidence frequency histogram (blue = correct answer, red = wrong answer). *Bottom sub-row*: calibration histogram with 5%-wide bins for visual resolution, where bar height represents observed accuracy within that bin; the dashed diagonal represents perfect calibration. Config 4 (Full System) consistently produces the most evenly spread confidence distributions and aligns most closely with the perfect-calibration diagonal across all datasets.

## 4.3 Discrimination Analysis

Figure 5 shows ROC curves for all four datasets. Config 4 achieves the highest AUROC on MedQA-250 (0.630, +0.056 over baseline) and MedMCQA-250 (0.594, +0.058). On MedQA-100, Config 2 achieves the highest AUROC (0.678, +0.141 over baseline), with Config 4 at 0.645 (+0.108). On MedMCQA-100, Config 3 achieves the highest AUROC (0.541) while Config 4 achieves 0.518. The ΔAUROC column in Table 1 further shows that Two-Phase Verification alone (C2) produces negative AUROC change on three of four datasets. The mechanism behind this pattern is examined in Section 5.3.

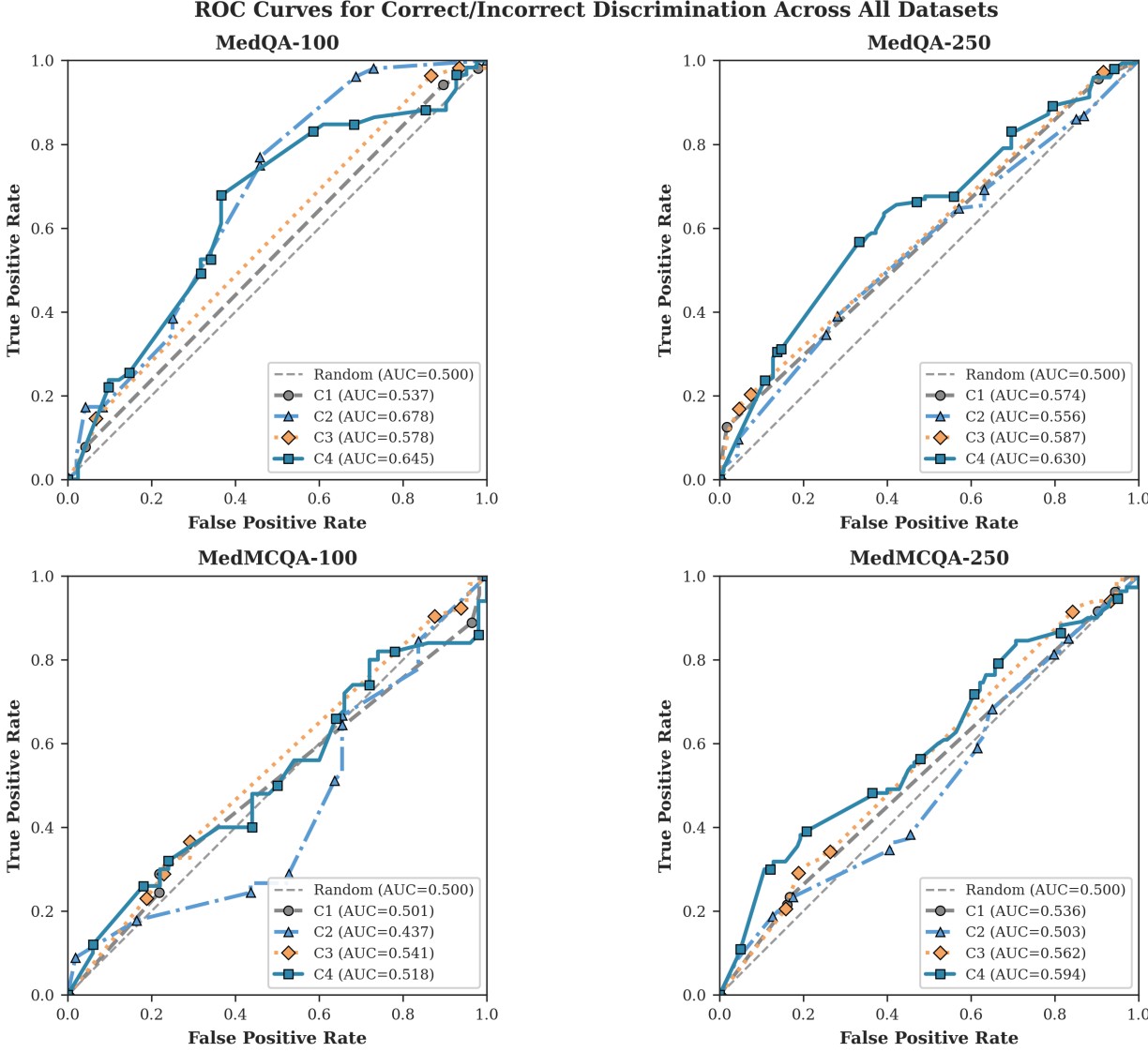

Figure 5: ROC curves for all four datasets. Each panel plots the four configuration curves with the random baseline (diagonal). Config 4 achieves the highest AUROC on MedQA-250 and MedMCQA-250. On MedQA-100, Config 2 (single specialist with verification) achieves the highest AUROC, and on MedMCQA-100, Config 3 (multi-agent without verification) is best, illustrating that discrimination benefits from multi-agent fusion scale more reliably with dataset size and question type.

### 4.4 Ablation Summary

Table 2 summarises the isolated contribution of each component on MedQA-250, the most reliable large-scale dataset. Figure 6 shows the same decomposition visually across all four datasets.

Table 2: Ablation summary on MedQA-250 ($n = 250$). Each row reports the *marginal* effect of adding one component.

| Component Added | $\Delta$Acc | $\Delta$ECE | $\Delta$AUROC |
|---|---|---|---|
| Two-Phase Verif. (C1→C2) | 0.0 pp | $-0.177$ $(-49.9\%)$ | $-0.018$ |
| Multi-Agent (C1→C3) | $+2.8$ pp | $-0.019$ $(-5.4\%)$ | $+0.013$ |
| Two-Phase added to Multi (C3→C4) | $+2.0$ pp | $-0.245$ $(-72.9\%)$ | $+0.043$ |
| Full System (C1→C4) | $+4.8$ pp | $-0.264$ $(-74.4\%)$ | $+0.056$ |

Multi-agent reasoning accounts for most of the accuracy gain ($+2.8$ pp, C1→C3), while Two-Phase Verification accounts for most of the calibration gain ($-49.9\%$ ECE, C1→C2; $-72.9\%$ ECE, C3→C4). Together they yield $+4.8$ pp accuracy and $-74.4\%$ ECE on MedQA-250, with the same qualitative pattern replicated across all four datasets.

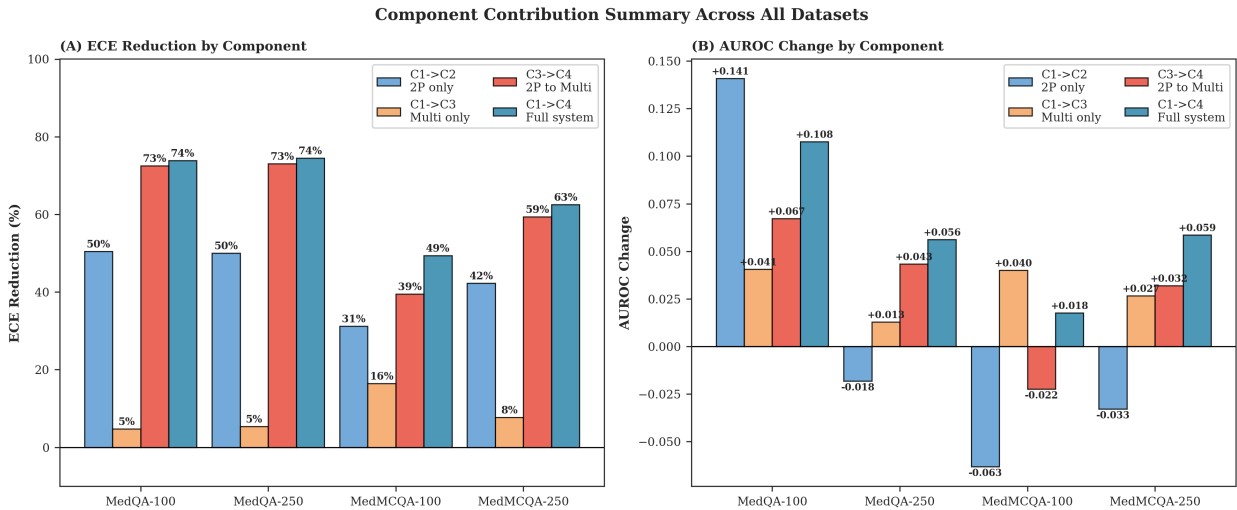

Figure 6: (A) ECE reduction percentage and (B) AUROC change for each component transition across all four datasets. Adding the full system (C1→C4, dark blue) consistently delivers the largest ECE reduction (49–74%) across all settings. Multi-agent reasoning (C1→C3, orange) drives the largest AUROC gains on MedQA subsets, while verification components dominate calibration improvement throughout.

## 5 Discussion

### 5.1 Multi-Agent Reasoning and Accuracy

High-disagreement questions are, by construction, cases where a single specialist is likely to err. Pooling four independently prompted specialists increases the chance that at least one agent reasons from the relevant domain, and majority voting suppresses idiosyncratic errors. S-Score Weighted Fusion sharpens this further, down-weighting agents whose verification step reveals internal contradictions so that the final answer is steered by the specialists whose reasoning is most self-consistent.

## 5.2 Two-Phase Verification and Calibration

The unverified configurations (C1, C3) produce heavily right-skewed confidence distributions with means near 0.90 (Figure 4), well above their actual accuracy of 43–57%. Two-Phase Verification brings these means down to roughly 0.55 by penalising confidence proportionally to measured inconsistency, making the confidence score a more faithful proxy for correctness, as confirmed by ECE reductions of 49–74% across all four datasets when Two-Phase Verification is combined with multi-agent fusion (C4). Crucially, this compression requires no ground-truth labels, as the signal comes entirely from the model's own reasoning.

## 5.3 Calibration, Discrimination, and the Limits of Verification

The $\Delta$AUROC column in Table 1 reveals a consistent pattern. Two-Phase Verification alone (C2) produces *negative* AUROC change on three of four datasets ($-0.018$ on MedQA-250, $-0.064$ on MedMCQA-100, $-0.033$ on MedMCQA-250). This is not a contradiction with the strong ECE improvements reported for C2. ECE measures whether confidence scores match observed accuracy on average, while AUROC measures whether their rank ordering correctly separates correct from incorrect predictions. These are different properties, and improving one does not guarantee improvement in the other.

Two-Phase Verification improves calibration by compressing all confidence scores toward the model's actual accuracy level, reducing systematic overconfidence. However, this compression is proportional to measured inconsistency, which is an imperfect proxy for whether an answer is actually right. A specialist that produces a self-consistent but wrong answer still receives a high S-score, while one whose reasoning is internally inconsistent but happens to be correct is penalized. The absolute values of confidence improve (lower ECE), but the rank ordering does not reliably improve alongside them, leaving AUROC unchanged or slightly reduced.

Multi-agent fusion restores and extends discrimination. When four independently prompted specialists vote and their S-scores weight the fusion, the correct answer tends to attract specialists with higher S-scores, though not because verification detects correctness directly, but because a correct answer is more likely to elicit self-consistent reasoning across diverse agents. The ensemble effect sharpens the separation between confident-correct and confident-incorrect predictions, which is what AUROC captures. The full system (C4) consequently achieves positive AUROC gains on all four datasets, with the largest improvements on the 250-question sets where the ensemble signal is more stable. The two components address different failure modes. Verification reduces overconfidence, multi-agent fusion improves ranking, and the full system needs both.

## 5.4 Benchmark Characteristics and Calibration Robustness

The full system achieves 59.2% on MedQA-250 but only 44.0% on MedMCQA-250 ($-15.2$ pp), a gap that persists at the 100-question scale (50.0% vs. 52.0% for Config 4 vs. Config 3 on MedMCQA-100). Both datasets use 4-option questions with a 25% random baseline, so the difference cannot be attributed to the number of choices.

The gap traces to differences in question structure. MedQA clinical vignettes average 109 words and present a full patient scenario, and the correct answer can often be reached by clinical reasoning even without memorizing the specific fact. MedMCQA stems average 15 words and test narrow factual recall from AIIMS PG and NEET-PG, covering anatomical minutiae, rare pharmacology, and pathology specifics across 21 specialties (Pal et al., 2022). At 7B parameters, the model does not retain sufficient domain-specific knowledge for many of these questions (Singh et al., 2025; Kadavath et al., 2022).

This knowledge gap interacts with verification in a specific way. On MedMCQA, moving from C3 to C4 (adding verification to multi-agent) *reduces* accuracy by 2.8 pp (250q) and 2.0 pp (100q) while still cutting ECE by 59% and 39%. A specialist can produce internally consistent reasoning that is nonetheless factually wrong. Verification then scores such reasoning highly and may override a correct majority vote (Wu et al., 2024). We discuss this limitation further in Section 5.5.

Despite that accuracy tension, the calibration gains hold. ECE is reduced by 49–74% in every configuration that includes Two-Phase Verification, across all four datasets. Even when the model lacks the knowledge to answer correctly, it can still be made to express appropriate uncertainty.

### 5.5 Limitations

The core tension in the framework is that consistency does not imply correctness. Two-Phase Verification measures internal coherence rather than factual accuracy, so a specialist that produces a self-consistent but wrong answer still receives a high S-score. This limitation is inherent in Wu et al.'s method and is sharpest on knowledge-recall benchmarks such as MedMCQA, where a model can reason coherently about a factually wrong premise.

The scope of the evaluation is also narrow. Both datasets were pre-filtered to four specialties (respiratory, cardiology, neurology, and gastroenterology), and it is not clear how well these findings would transfer to anatomy, pharmacology, surgery, or the many other domains covered by medical licensing examinations. A more specific form of this limitation appears in the MedMCQA results. The four specialist agents were configured for their respective domains, yet MedMCQA spans 21 specialties including many that none of the specialists were designed to handle. Questions from these unrepresented areas are evaluated by agents whose domain prompts do not match the question content, which likely depresses both accuracy and the reliability of S-scores as a proxy for domain competence. Attention-based fusion that learns per-specialty reliability weights from a small validation set would be a natural extension to address this domain mismatch.

Using a dedicated verifier model separate from the specialist agents would eliminate the risk of a specialist confirming its own reasoning errors through self-verification. This design change is outside the scope of the current single-GPU setup. Simultaneously loading a specialist and a separate verifier would require substantially more GPU memory than the 24 GB available on the RTX 5090 used here, making it a direction that depends on expanded infrastructure rather than algorithmic adjustment.

Computational cost is a practical concern for deployment. Config 4 requires approximately $16\times$ the LLM calls of Config 1, because each of the four specialists requires four calls covering the initial answer and three verification sub-calls for question formulation, independent answering, and reference answering. On a single NVIDIA RTX 5090 GPU this amounts to 833 minutes for MedQA-250 under Config 4 versus 116 minutes under Config 1 ($7.2\times$ wall-clock). For MedMCQA-250 the figures are 712 versus 102 minutes ($7.0\times$). Running the four specialists in parallel is the most direct way to reduce this overhead.

## 6 Future Work

The most consequential open problem is the one identified in Section 5.5. A specialist can produce self-consistent but factually wrong reasoning, and the verification step has no way to catch it. Retrieval augmentation is the highest-priority direction forward. Grounding Phase 2 verification against a curated medical knowledge base such as PubMed or clinical practice guidelines would replace pure internal consistency with evidence-anchored consistency, allowing the S-score to penalize claims that contradict established medical knowledge rather than only claims that contradict the specialist's own prior reasoning. This addresses the core limitation directly, without requiring additional model scale or training data, and is therefore the most tractable near-term extension.

A natural comparison left unmeasured in this work is against self-consistency sampling (Wang et al., 2023), which generates multiple reasoning chains from a single model and selects the most frequent answer. At matched inference-time compute, it is not clear whether self-consistency or MARC's verification approach yields better calibration. A compute-controlled comparison would clarify whether the gains reported here reflect the multi-agent structure, the verification step, or primarily the additional inference budget.

The evaluation is restricted to high-disagreement subsets of two benchmarks. Extending to the full MedQA and MedMCQA distributions would test whether calibration gains persist on questions where the specialists tend to agree and uncertainty quantification is less critical, providing a more complete picture of practical deployment performance.

The current design includes two hyperparameters that were set without systematic search. The Jaccard consistency threshold $\tau = 0.4$ determines whether a verification question pair is counted as consistent or not, and the number of verification questions per specialist is fixed at four. A sensitivity analysis varying these parameters would clarify how robust the calibration gains are to these choices, which is particularly important given the small evaluation sets used here.

Whether the relative benefit of Two-Phase Verification persists as model scale increases is an open empirical question. Larger models are better calibrated to begin with (Kadavath et al., 2022), so absolute ECE gains may shrink. If the relative reduction as a fraction of baseline ECE remains stable, verification retains its value at larger scales. If it diminishes, this would suggest that verification is most useful as a compensatory mechanism for smaller models rather than a general-purpose calibration tool, and would help clarify where in the model size spectrum the framework adds the most value.

A separate open question is whether the fusion weights can be improved with learned parameters. A small specialist performance validation set could provide signal for per-specialty reliability that the current heuristic formulation ignores, and attention-based fusion mechanisms represent a promising direction for addressing the domain mismatch identified in Section 5.5.

## 7   Conclusion

This paper examined whether combining domain-specific specialist agents with consistency-based self-verification improves uncertainty calibration in medical multiple-choice QA evaluated on high-disagreement subsets of MedQA-USMLE and MedMCQA. ECE falls by 49–74% across all four evaluation settings relative to the single-specialist baseline, a gain that holds on both benchmarks. These gains persist when the baseline is taken as the average across all four individual specialists rather than the respiratory agent alone (Table 4). Ablation analysis traces this improvement primarily to Two-Phase Verification, while multi-agent reasoning contributes most of the accuracy gains. Together, the full system achieves ECE = 0.091 on MedQA-250 (74.4% reduction) and improves AUROC by up to +0.108 on MedQA-100.

A key empirical observation is that calibration and discrimination respond to distinct mechanisms within the framework. Two-Phase Verification compresses overconfident scores toward the model's actual accuracy level, reducing systematic miscalibration without necessarily improving the rank ordering of predictions. Multi-agent fusion restores and extends discrimination by weighting the ensemble toward specialists whose reasoning is most self-consistent. The full system requires both components, and the ablation confirms that neither alone reproduces the combined effect. This decomposition may generalize to other domains where inference-time uncertainty quantification without labeled calibration data is needed.

The MedMCQA results are worth emphasizing. Even when absolute accuracy stagnates because the 7B model lacks the specialised factual knowledge the benchmark demands, the calibration mechanism keeps working, assigning lower confidence to the answers the system gets wrong. A model too limited to answer reliably can still be made to express appropriate uncertainty, which may provide a useful signal for routing difficult questions to human review. Whether this signal is sufficient to guide clinical deferral decisions in practice is a question that the controlled evaluation setting studied here cannot fully answer, and it remains a direction for future investigation.

What remains unsolved is the confidence the system assigns to answers that are internally consistent but factually incorrect. Connecting the verification step to an external medical knowledge source, such that Phase 2 checks claims against established evidence rather than the specialist's own reasoning, is the most promising direction we see for future work.

### Broader Impact Statement

The system studied here is a research prototype and should not be used for actual clinical decisions without extensive prospective validation. The accuracy levels achieved (44–59%) are well below what autonomous clinical decision support would require. Better-calibrated confidence scores could help clinicians identify cases where AI recommendations warrant additional scrutiny, but this benefit depends on the calibration holding

in deployment settings not covered by our evaluation. The specialist agents' knowledge is bounded by the pretraining data of a 7B model, which may reflect geographic, demographic, and linguistic biases in that corpus.

**Acknowledgments**

[Anonymised for review.]

**Data and Code Availability**

Evaluation datasets (MedQA-USMLE, MedMCQA) are publicly available. Experimental code and result files will be released upon publication at `https://github.com/[anonymised]`.

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

# A  Appendix

## A.1  Dataset Construction

High-disagreement subsets are constructed by the following three-step procedure applied identically to both benchmarks:

1. **Specialist curation.** A lightweight curation pass runs all four specialist agents on the full source datasets (approximately 10,000 questions for MedQA and 49,000 for MedMCQA). Each agent is prompted to respond with only the option letter (A/B/C/D) and no explanation (`max_new_tokens=8`), since only the letter is needed to detect disagreement and this imposes no quality loss relative to the main experiments. Early stopping halts after collecting 220 high-disagreement and 60 agreement candidates.

2. **Disagreement labeling.** A question is labeled high-disagreement if at least two of the four specialists propose different answers.

3. **Subset construction.** From the curated pool, 100-question and 250-question subsets are drawn by taking the top-ranked high-disagreement questions (sorted by number of differing specialist answers) and a random sample of agreement questions (fixed seed for reproducibility), retaining only questions with valid ground-truth labels ($cop \in \{1, 2, 3, 4\}$ for MedMCQA; `answer_idx` present for MedQA). The combined set is shuffled before evaluation.

## A.2 Prompt Templates

**Specialist Agent — System Prompt**

```
You are an expert medical specialist in {specialty}.
You have deep knowledge in this specific domain and are
consulting on a medical case.

{knowledge_context}

Your task is to:
1. Analyze the question from your specialty's perspective
2. Provide your expert opinion on the correct answer
3. Explain your reasoning clearly
4. Include relevant medical knowledge from your specialty

Be precise, evidence-based, and focus on your area of expertise.
```

**Specialist Agent — User Prompt**

```
Question: {question}

Options:
{options}

As a {specialty} specialist, please use CHAIN-OF-THOUGHT reasoning:

STEP 1: Understand the clinical scenario
- What are the key symptoms, signs, or findings?
- What is the patient's presentation?
- What is the clinical context?

STEP 2: Consider differential diagnoses from your specialty
- What conditions in your specialty could explain this?
- What are the diagnostic criteria for each?
- What are the distinguishing features?

STEP 3: Evaluate each option systematically
- For EACH option, evaluate:
  * Is this medically correct?
  * Does this fit the clinical scenario?
  * What evidence supports or refutes this?

STEP 4: Compare options
- Which option best fits the clinical scenario?
- Which option has the strongest evidence?
```

STEP 5: Make your decision
- Select the most appropriate answer
- Provide confidence score (0-1)

IMPORTANT: For ANSWER, provide ONLY the option LETTER
(A, B, C, or D). Do NOT provide the full text.

Format your response as:
STEP_1_ANALYSIS: [clinical scenario analysis]
STEP_2_DIFFERENTIAL: [differential diagnoses]
STEP_3_OPTION_EVALUATION: [evaluation of each option]
STEP_4_COMPARISON: [comparison of options]
STEP_5_DECISION: [final decision reasoning]
ANSWER: [Single letter: A, B, C, or D]
CONFIDENCE: [0.0-1.0]
REASONING: [detailed explanation]

**Two-Phase Verification — Question Formulation Prompt**

You are a medical verification expert. Your task is to
formulate verification questions based on the explanation.

Question: {question}
Proposed Answer: {answer}
Explanation: {reasoning}

Based on the explanation above, formulate exactly 4 specific
verification questions that check the factual claims made in
the explanation. These questions should:
1. Target specific medical facts or claims in the explanation
2. Be answerable independently (without the explanation)
3. Help verify the correctness of the reasoning
4. Cover the key factual claims in the explanation

Format your response as:
VERIFICATION_QUESTIONS:
1. [First question]
2. [Second question]
3. [Third question]
4. [Fourth question]

**Two-Phase Verification — Independent Answering Prompt**

You are a medical expert. Answer the following verification
questions based on your medical knowledge.

Original Question Context: {question}

Verification Questions:
{verification_questions}

IMPORTANT: Answer these questions based on your medical
knowledge ONLY. Do NOT reference any specific explanation
or reasoning from the original question.

```
Format your response as:
ANSWERS:
1. [Answer to first question]
2. [Answer to second question]
3. [Answer to third question]
4. [Answer to fourth question]
```

**Two-Phase Verification — Reference Answering Prompt**

```
You are a medical expert. Answer the following verification
questions while referencing the provided explanation.

Original Question Context: {question}

Original Explanation:
{reasoning}

Verification Questions:
{verification_questions}

IMPORTANT: Answer these questions while referencing the
explanation above. Use the explanation to guide your answers.

Format your response as:
ANSWERS:
1. [Answer to first question]
2. [Answer to second question]
3. [Answer to third question]
4. [Answer to fourth question]
```

### A.3 Bootstrap Confidence Intervals

Table 3 reports 95% bootstrap confidence intervals ($n$=10,000 resamples, percentile method, fixed seed) for all three metrics, computed by resampling individual question outcomes, and covers the four ablation configurations (C1–C4). Point estimates for all metrics appear in Table 1. The *Avg. 4 Specialists* row in Table 1 is a post-hoc descriptive statistic across independent specialist runs and is not a runnable configuration, so no bootstrap CI is reported for it. Per-specialist results appear in Table 4.

On three of four datasets (MedQA-100, MedQA-250, MedMCQA-250) the C4 ECE interval lies entirely below the C1 interval, indicating the calibration improvement is unlikely to be a sampling artifact. On MedMCQA-100 the ECE intervals partially overlap, which is unsurprising at $n$=100. Accuracy and AUROC intervals overlap across all settings. The accuracy and discrimination gains reported in Table 1 should be read as directional rather than conclusive at these sample sizes.

### A.4 Per-Specialist Baseline Performance

Table 4 reports accuracy, ECE, and AUROC for each of the four specialist agents running independently across all evaluation sets. Each specialist uses the same prompt and model as in the main experiments, but receives no verification step and no fusion. The Respiratory specialist corresponds exactly to the Config 1 baseline in Table 1. The Average row is the mean across all four specialists and represents a stronger single-specialist baseline than Respiratory alone. By ECE, Respiratory has the highest (worst) value on two of four datasets (MedQA-250 and MedMCQA-100), is tied for highest with Cardiology on a third (MedMCQA-250 at 0.469), and is second-highest on the fourth (MedQA-100 at 0.374, against Gastroenterology's 0.380). Respiratory is therefore among the worst-calibrated specialists on every evaluation set. The improvements of Config 4 over

Table 3: Bootstrap confidence intervals (95%, $n$=10,000 resamples) for all metrics across all four datasets and configurations (C1–C4).

| Dataset | Config | **Accuracy** 95% CI | **ECE** 95% CI | **AUROC** 95% CI |
|---|---|---|---|---|
| MedQA-100 | C1: Single Specialist | [42.0, 62.0] | [0.269, 0.463] | [0.470, 0.605] |
| | C2: Single + Two-Phase | [42.0, 62.0] | [0.103, 0.272] | [0.572, 0.781] |
| | C3: Multi + S-Score (No 2P) | [45.0, 65.0] | [0.244, 0.438] | [0.502, 0.652] |
| | C4: Full System | [49.0, 69.0] | [0.050, 0.194] | [0.531, 0.755] |
| MedQA-250 | C1: Single Specialist | [48.4, 60.4] | [0.291, 0.412] | [0.533, 0.616] |
| | C2: Single + Two-Phase | [48.4, 60.4] | [0.133, 0.245] | [0.486, 0.626] |
| | C3: Multi + S-Score (No 2P) | [51.2, 63.2] | [0.269, 0.388] | [0.539, 0.634] |
| | C4: Full System | [53.2, 65.2] | [0.064, 0.159] | [0.561, 0.699] |
| MedMCQA-100 | C1: Single Specialist | [35.0, 55.0] | [0.257, 0.443] | [0.406, 0.598] |
| | C2: Single + Two-Phase | [35.0, 55.0] | [0.234, 0.415] | [0.323, 0.552] |
| | C3: Multi + S-Score (No 2P) | [42.0, 62.0] | [0.217, 0.400] | [0.438, 0.643] |
| | C4: Full System | [40.0, 60.0] | [0.140, 0.312] | [0.404, 0.632] |
| MedMCQA-250 | C1: Single Specialist | [36.8, 48.8] | [0.319, 0.439] | [0.477, 0.595] |
| | C2: Single + Two-Phase | [36.8, 48.8] | [0.201, 0.315] | [0.431, 0.575] |
| | C3: Multi + S-Score (No 2P) | [40.8, 52.8] | [0.298, 0.415] | [0.497, 0.626] |
| | C4: Full System | [37.6, 50.0] | [0.123, 0.226] | [0.521, 0.666] |

both the Respiratory baseline and the average single-specialist baseline remain consistent and substantial across all four evaluation sets. Config 4 also achieves lower ECE than every individual specialist across all four datasets, including the best-calibrated one on each dataset, confirming that the calibration gains are not an artifact of a weak single-specialist reference point.

Table 4: Per-specialist performance (Accuracy, ECE, AUROC) across all four evaluation sets. Each specialist runs independently with no verification and no fusion (equivalent to Config 1 setup). The *Average* row is the mean across all four specialists and serves as an alternative single-specialist baseline. Respiratory corresponds to the Config 1 baseline in Table 1.

| | *MedQA-100* | | | *MedQA-250* | | | *MedMCQA-100* | | | *MedMCQA-250* | | |
|---|---|---|---|---|---|---|---|---|---|---|---|---|
| Specialist | Acc. | ECE | AUROC | Acc. | ECE | AUROC | Acc. | ECE | AUROC | Acc. | ECE | AUROC |
| Respiratory | 52.0% | 0.374 | 0.537 | 54.4% | 0.355 | 0.574 | 45.0% | 0.469 | 0.501 | 42.8% | 0.469 | 0.536 |
| Cardiology | 57.0% | 0.343 | 0.541 | 57.6% | 0.322 | 0.537 | 47.0% | 0.451 | 0.557 | 42.8% | 0.469 | 0.562 |
| Neurology | 55.0% | 0.343 | 0.587 | 58.4% | 0.309 | 0.555 | 48.0% | 0.418 | 0.514 | 44.8% | 0.448 | 0.543 |
| Gastroenterology | 53.0% | 0.380 | 0.543 | 58.4% | 0.315 | 0.552 | 51.0% | 0.368 | 0.589 | 48.4% | 0.399 | 0.570 |
| *Average* | *54.2%* | *0.360* | *0.552* | *57.2%* | *0.325* | *0.555* | *47.8%* | *0.426* | *0.540* | *44.7%* | *0.446* | *0.553* |

