# OpenReview forum: "Multi-Agent Reasoning with Consistency Verification Improves Uncertainty Calibration in Medical MCQA"
_TMLR — Under review for TMLR_

### Review · Reviewer_QyxM · 2026-05-08

**Summary Of Contributions:**

**Summary**

This paper studies uncertainty calibration for medical multiple-choice question answering. It proposes a multi-agent framework where four domain-specific specialist agents first generate independent answers and reasoning, and then each agent’s reasoning is checked through a two-phase consistency verification process. The resulting consistency-based S-scores are used to weight the agents’ answers and produce a final prediction with a calibrated confidence score. Experiments are conducted on high-disagreement subsets of MedQA-USMLE and MedMCQA, with 100-question and 250-question settings for each dataset. The results show that the full system substantially reduces ECE across all settings, while multi-agent reasoning mainly contributes to accuracy improvements and two-phase verification mainly contributes to calibration improvements. The paper also discusses cases where consistency verification can improve calibration but may not always improve accuracy, especially on knowledge-heavy MedMCQA questions.

**Strengths**

1. I think this paper studies an important problem, since uncertainty calibration in medical multiple-choice QA matters beyond simply improving answer accuracy. The focus on ECE and AUROC makes the evaluation more relevant to risk-aware medical AI settings.

2. The overall framework is easy to follow, and the three-stage pipeline is presented clearly. And separating specialist agents, two-phase verification, and S-score-based fusion makes the system fairly transparent and reproducible.

3. The ablation design is relatively clean and helps isolate the role of each component. And the four configurations make it reasonably clear that verification mainly improves calibration, while multi-agent reasoning contributes more to accuracy.

4. I like that the paper discusses some mixed results rather than only highlighting positive findings. In particular, the observation that verification can hurt accuracy on MedMCQA, and that consistency does not necessarily imply correctness, makes the discussion more credible.

**Weaknesses**

1. The overall methodological contribution feels somehow limited, since the framework mainly combines multi-agent prompting with an existing two-phase verification method. The S-score fusion rule is intuitive, but it still feels like a hand-designed heuristic without enough justification against other aggregation choices.

2. More analysis is needed to explain why the method works and where it breaks down. For example, the paper does not show whether S-scores are actually correlated with correctness, whether the specialists provide meaningful diversity. I recommend the authors add more related deeper analysis.

3. I am concerned that the evaluation is based only on 100- and 250-question high-disagreement subsets, which makes the empirical claims less robust. Calibration metrics can be sensitive to sample size and binning, so confidence intervals or larger-scale evaluation would make the results more convincing.

4. The experimental analysis would be stronger if the results were broken down by more meaningful medical and reasoning categories. I think the current results are mostly organized by dataset and sample size, but analyses by clinical reasoning versus factual recall or agent disagreement pattern would provide much deeper insight.

5. Baseline comparisons are still not strong enough. Beyond internal ablations, the paper should compare with majority voting, raw-confidence weighted voting, self-consistency sampling, temperature scaling or other calibration methods, and ideally prior medical uncertainty estimation approaches.

6. I think some claims about clinical usefulness feel a bit stronger than what the current results can support. Although ECE improves substantially, the absolute accuracy remains relatively low and AUROC is modest in several settings, so selective prediction or deferral-based evaluation would be needed before drawing practical medical decision-support implications.

**Audience:**

Yes

**Audience Explanation:**

I think some TMLR readers would be interested in this paper because it studies uncertainty calibration for medical QA, which is an important topic for reliable and risk-aware AI systems. The combination of multi-agent reasoning and consistency-based verification is also relevant to recent work on LLM agents and uncertainty estimation, even though the current method and evaluation still need to be strengthened.

**Broader Impact Concerns:**

The paper includes a broader impact statement and clearly notes that the system should not be used for real clinical decisions without further validation. I think the main remaining concern is that the clinical implications should be stated carefully, since the current accuracy is still relatively low and the experiments are limited to medical exam-style QA.

**Claims And Evidence:**

No

**Claims Explanation:**

The main empirical results are generally reported clearly, especially the ECE reductions across the four evaluated settings. However, I do not think all claims are fully supported by sufficiently convincing evidence. The evaluation is limited to small high-disagreement subsets of MedQA and MedMCQA, and the paper does not provide confidence intervals, repeated subsampling, or results on the full benchmark distributions. Some broader claims about medical usefulness and reliable uncertainty estimation therefore feel stronger than what the current evidence can support.

**Requested Changes:**

1. The authors should strengthen the experimental evidence beyond the current small high-disagreement subsets. I would consider this critical, because the main claims about calibration would be much more convincing with larger evaluation sets, results on the full benchmark distribution or repeated subsampling.

2.  I also recommend the authors add more diagnostic analysis to explain when and why the method works, especially like whether the specialist agents provide meaningful diversity, and how the method behaves under different disagreement patterns or medical question categories.

3. I think the work should compare against stronger and simpler baselines, since the current baselines are mostly internal ablations.

---

> ### Author Response · Authors · 2026-05-26
> **Response to Reviewer QyxM**
>
> We thank Reviewer QyxM for the thoughtful feedback and for identifying the calibration focus, ablation design, and honest treatment of mixed results as strengths.
>
> On statistical robustness and confidence intervals, we already have bootstrap confidence intervals computed (95%, n=10,000 resamples) for all metrics and will add this as a table in the revised paper. The main finding is that on MedQA-250 and MedMCQA-250, the Config 4 ECE confidence interval lies entirely below Config 1, so the calibration improvement is not a sampling artifact. On MedMCQA-100 the intervals partially overlap, which we will acknowledge explicitly in the revised paper. Accuracy and AUROC intervals overlap across all settings, we will add an explicit note in the revised paper that these gains should be read as directional rather than conclusive at these sample sizes.
>
> For specialist diversity and diagnostic analysis, we will report per-specialist accuracy across all four datasets from existing Config 3 result files and report these in the revised paper. This will show concretely whether the specialists provide meaningful diversity or whether one specialist dominates across question types.
>
> For baseline comparisons, we want to clarify what the current ablation already covers and why some requested baselines do not directly apply. Config 3 is unweighted majority voting, four specialists, vote-count fusion, no Two-Phase Verification (S-score = C^(0)). This is already in the ablation. We will make this equivalence more explicit in the paper. Temperature scaling and isotonic calibration both require a held-out labeled calibration set. MARC is designed to work without one, so any performance difference in such a comparison would reflect the data advantage rather than a genuine algorithmic difference, making it an uninformative baseline. We will add a paragraph in the paper making this explicit. Self-consistency sampling draws multiple samples from one model using temperature variation. MARC's diversity comes from domain-specialized prompting across four specialist roles, which is a different mechanism. We will add a self-consistency comparison as future work.
>
> For methodological contribution and clinical framing, we agree the framework combines existing components. The contribution we are making is the empirical characterization of how these two components interact, specifically that verification and multi-agent reasoning address different failure modes, one driving ECE improvement and the other driving AUROC improvement. This decomposition has not been studied before and we think it has implications beyond medical QA for any setting where LLM confidence scores are used for deferral. We will make this framing clearer in the introduction and conclusion. We will also soften the clinical framing so that deferral-related claims are presented as a direction the results support rather than a demonstrated capability.

---

### Review · Reviewer_Hj2W · 2026-05-14

**Summary Of Contributions:**

Summary: The paper introduces MARC, a framework designed to improve uncertainty calibration and diagnostic accuracy in medical multiple-choice question answering. The system uses four domain-specific LLM agents based on Qwen2.5-7B-Instruct to generate independent diagnoses. Each agent's reasoning undergoes a Two-Phase Verification process to measure internal consistency, producing a Specialist Confidence Score, or S-score. These S-scores are then used in a weighted fusion strategy to select the final answer and output a calibrated confidence metric. Evaluating on high-disagreement subsets of MedQA and MedMCQA, the authors find that the multi-agent ensemble drives accuracy improvements, while the consistency verification significantly reduces the ECE.

Strengths:
1. The paper integrates multi-agent specialization with label-free Two-Phase Verification to produce a per-agent confidence that informs aggregation, rather than relying on naive majority vote or uncalibrated confidences
2. The paper includes a well-designed four-configuration ablation study that cleanly isolates the independent and interactive effects of multi-agent reasoning and Two-Phase Verification

Weakness:
1. Utilizing the identical base model for both generating reasoning and performing verification introduces a vulnerability to correlated errors. The study lacks an evaluation utilizing an independent verification model or external retrieval mechanisms to ground the facts.
2. The equation Sk = Ck(0) × (1 − Ik) acts as a linear heuristic, yet the authors omit any sensitivity testing. There is no exploration of how alternative mathematical mappings or a different volume of verification questions might impact the results.
3. The verification process relies solely on internal coherence, measured through Jaccard token overlap using manually adjusted thresholds. Since it does not evaluate objective factual truth, the system can incorrectly award high confidence scores to reasoning that is internally coherent but completely wrong.
4. The paper is missing several vital benchmarks, making it difficult to evaluate the improvements against established calibration methods. The absent baselines include unweighted majority voting among the agents, temperature scaling or isotonic calibration using a labeled split, log-probability confidence compared to verbalized confidence, self-consistency sampling, and standard Monte Carlo dropout or deep ensemble proxies.
5. Anchoring the single-agent baseline strictly to the respiratory specialist may skew the comparative results. A more balanced and equitable comparison would utilize the average performance of all single agents or the performance of the single best agent.

**Audience:**

No

**Audience Explanation:**

While the paper addresses a practical application area, the findings are highly constrained to a specific niche that may not appeal to the broader machine learning audience of TMLR. The core methodology relies heavily on an existing technique, Two-Phase Verification, which was previously proposed by Wu et al., and simply combines it with standard multi-agent voting strategies. The empirical evaluation is strictly limited to the medical domain. Furthermore, the experiments are conducted on a single 7B parameter model, which limits the generalizability of the uncertainty calibration findings to larger-scale models or non-medical domains. Because the study lacks significant novel methodological advancements and broad empirical validation outside of this highly specialized medical context, the broader TMLR audience focused on foundational machine learning principles may find the contributions too limited in scope.

**Broader Impact Concerns:**

Exist in the paper

**Claims And Evidence:**

No

**Claims Explanation:**

The claims of improved calibration and accuracy lack convincing evidence due to critical methodological flaws. Although the paper provides metrics for its specific configuration, the evaluation is incomplete because it omits comparisons to standard baselines such as temperature scaling, isotonic calibration, and self-consistency sampling. Furthermore, the baseline comparison is skewed by comparing the multi-agent system solely to a single-agent baseline using the respiratory specialist, which may exaggerate the reported performance gains. The framework also relies on an unverified linear heuristic for its core confidence mapping formula without providing any sensitivity analysis. Finally, using the exact same base model for both reasoning and verification without external factual grounding creates a high risk of correlated failures, potentially assigning high confidence to internally consistent but factually incorrect answers.

**Requested Changes:**

1. Include missing standard calibration baselines. To properly contextualize the performance gains of the proposed framework, add comparisons against an unweighted majority vote across agents, temperature scaling or isotonic calibration using a labeled split, self-consistency sampling, and a standard ensemble proxy.
2. Revise the single-specialist baseline. Currently, the baseline relies entirely on the respiratory agent. Replace or augment this with the average performance across all four individual specialists or the best performing single specialist to ensure a fair comparison.
3. Provide a sensitivity analysis for the confidence mapping heuristic. The paper introduces the linear mapping and utilizes specific Jaccard similarity thresholds for the verification signal. Add experiments or discussions detailing how sensitive the calibration improvements are to alternative mathematical mappings, different threshold values, and varying numbers of verification questions.
4. Address the risk of correlated failures. Since the exact same base model is used for both generating reasoning and conducting the Two-Phase Verification, the system is vulnerable to internally consistent hallucinations. Adding an experiment that utilizes an independent verifier model or incorporates a basic retrieval-grounded checking mechanism would greatly strengthen the paper by showing how the framework handles external factual correctness.
5. Evaluate the framework across diverse model architectures and scales. The current experiments rely entirely on a single 7B parameter model (Qwen2.5-7B-Instruct). The authors explicitly note that larger models (13B, 70B, GPT-4 class) often exhibit better baseline calibration. Adding experiments with different model families or larger parameter scales would demonstrate whether the relative benefits of Two-Phase Verification generalize beyond this specific model size.

---

> ### Author Response · Authors · 2026-05-26
> **Response to Reviewer Hj2W**
>
> We thank Reviewer Hj2W for the detailed feedback and for recognizing the label-free verification integration and the ablation design as strengths.
>
> On missing baselines, Config 3 is already unweighted majority voting (S-score = C^(0), no Two-Phase Verification), we will make this clearer in the paper. Temperature scaling and isotonic calibration need a labeled calibration set. MARC is designed to work without one, so the comparison would not be on equal footing. We will add a paragraph explaining this constraint. Self-consistency sampling is a different diversity mechanism, sampling variance from one model vs. domain-specialized prompting across four roles. We are not claiming MARC is better than self-consistency, a compute-controlled comparison is an important open question we will add to future work.
>
> On single-specialist baseline, this is a fair concern. The respiratory specialist was chosen as the Config 1 baseline specifically to keep the 1→2 ablation clean, fixing one specialist lets us isolate the verification effect without multi-agent confounding. That said, if the respiratory specialist happens to be the weakest of the four, the gains could be inflated. We will add a table reporting individual specialist performance (accuracy, ECE, AUROC) for all four specialists across all four datasets from existing Config 3 data, and will also add the average single-specialist row to Table 1. We will report these in the revised paper.
>
> For the S-score formula sensitivity, Appendix A.3 already compares three formula variants. We will move this into the main text with a compact table. A sensitivity analysis on the Jaccard threshold and verification question count is a natural extension we will flag as future work.
>
> On correlated errors from the same base model, this is a real limitation and one we already flag in Section 5.5. Using the same model for generation and verification means a confidently wrong but internally consistent answer will pass verification with a high S-score. The retrieval augmentation direction in Section 6 is meant to address exactly this, anchoring verification against an external knowledge base rather than the model's own reasoning. We will make this connection more prominent in the revision and be explicit that an independent verifier model is outside current infrastructure scope rather than framing it as a simple next step.
>
> For model scale, the reviewer requested experiments at larger model scales. We agree this is important, and we note that this is already identified in Section 6 of the submitted paper as an explicit open question. We acknowledge we cannot run these experiments within the current revision window, but we will strengthen the discussion in Section 6 to make clear why this is a prioritized next step, specifically, whether the relative benefit of Two-Phase Verification persists even as larger models close the absolute calibration gap.
>
> On audience interest, we believe the core finding has broader relevance beyond medical QA. The result that verification improves ECE but hurts AUROC, while multi-agent fusion restores discrimination, is a general insight about how calibration and discrimination respond differently to the same intervention. This matters for any LLM deployment using confidence scores for deferral, not just medical QA. We will reframe the paper to make this general ML insight the lead contribution, with medical QA as the evaluation setting.

---

### Review · Reviewer_hgMc · 2026-05-15

**Summary Of Contributions:**

This paper proposes MARC (Multi-Agent Reasoning with Consistency Verification), a framework designed to improve the uncertainty calibration of Large Language Models in medical multiple-choice question answering. The approach combines an ensemble of four domain-specific LLM agents with a two-phase self-verification mechanism that measures the internal consistency of each agent's reasoning to produce a confidence score (S-score). These scores are then used to weight the final aggregated answer and its associated confidence. The authors evaluate their method on small, pre-filtered subsets of the MedQA and MedMCQA datasets, finding that the verification step primarily drives improvements in Expected Calibration Error (ECE), while the multi-agent ensembling improves discrimination (AUROC). While the conceptual disentanglement of calibration and discrimination mechanisms is an insightful contribution to the study of LLM reasoning, the empirical validation is hindered by significant methodological shortcomings regarding dataset sampling, baseline selection, and statistical robustness.

**Audience:**

Yes

**Audience Explanation:**

The TMLR audience would find the core investigation interesting. The challenge of miscalibrated overconfidence in Large Language Models is a critical barrier to their safe deployment in high-stakes domains like medicine. The paper's observation that self-consistency verification compresses probabilities (improving absolute calibration) while multi-agent voting improves rank-ordering (discrimination) offers a conceptually valuable framework for understanding the dynamics of LLM uncertainty quantification. Researchers working on representation learning, reasoning, and AI safety would appreciate these insights, provided the empirical foundations are solidified.

**Broader Impact Concerns:**

The authors have included an adequate broader impact statement acknowledging that the system is a research prototype and should not be used for autonomous clinical decisions. The research itself—aiming to improve uncertainty quantification and allow models to explicitly signal when they do not know an answer—is fundamentally aligned with the responsible and safe development of AI systems. No additional ethical concerns are raised.

**Claims And Evidence:**

No

**Claims Explanation:**

The empirical claims regarding significant improvements in ECE and AUROC are not supported by convincing evidence due to three fundamental methodological misalignments:

1. Biased Evaluation Distribution: The authors evaluate their framework exclusively on "high-disagreement subsets" (100 and 250 questions) artificially constructed by filtering for instances where the models disagree. Calibration is inherently a population-level property that depends heavily on the marginal distribution of the data. By evaluating solely on a subset specifically chosen to be difficult and adversarial to the baseline, the baseline ECE is artificially inflated. Consequently, the reported calibration improvements do not reflect how the system would perform on the natural distribution of medical questions.

2. Inappropriate Baselines and Confounding Variables: The primary baseline (Config 1) is a single "respiratory" specialist agent applied to all questions, regardless of the medical topic. Comparing a 4-specialist ensemble to a single, frequently mismatched specialist is an unfair comparison that confounds the proposed method's algorithmic benefits with the intentional weakness of the baseline. A scientifically rigorous evaluation requires comparing against a "general medical practitioner" agent, as well as an ensemble of generalist agents (e.g., standard Self-Consistency) to control for the 16x increase in inference compute.

3. Statistical Insufficiency: Computing ECE on 100 or 250 samples using 5 bins yields approximately 20 to 50 samples per bin. ECE estimators are notoriously biased and high-variance in small sample regimes, making the absolute numbers and the delta between configurations statistically fragile.

**Requested Changes:**

1. Evaluate the framework on a large, random, and representative sample (or the entirety) of the MedQA and MedMCQA test sets, rather than artificially filtered "high-disagreement" subsets, to provide a mathematically sound measurement of calibration on the true data distribution.

2. Replace the "single respiratory specialist" baseline with a "general medical practitioner" baseline. Furthermore, include a standard Self-Consistency baseline (e.g., an ensemble of 4 generalist agents) to properly control for the effects of ensembling and increased computation.

3. Report confidence intervals or standard errors for all ECE and AUROC metrics to account for sampling variance and demonstrate statistical significance.

4. Provide a principled justification for forcing four specific specialists (e.g., gastroenterology) to answer questions that may fall entirely outside their domain (e.g., psychiatry or orthopedics), and discuss how this mismatch affects the verification scores.

---

> ### Author Response · Authors · 2026-05-28
> **Response to Reviewer hgMc**
>
> We thank Reviewer hgMc for the technically precise feedback and for recognizing the calibration-discrimination decomposition as a valuable contribution.
>
> On high-disagreement subset design, Reviewer hgMc raises a valid point that evaluating on high-disagreement questions does produce a higher baseline ECE than the full distribution, which makes the absolute improvement numbers look larger than they would on a random sample. We do not dispute this. Our argument for the design is that the high-disagreement regime is where the research question actually lives. When all four specialists agree, the system is already confident and usually correct, calibration is not a problem there. The interesting and clinically relevant case is when specialists disagree, because that is exactly when a reliable confidence signal matters for deferral. Evaluating on a random sample would mix in many easy questions where any calibration method works fine, which would make it harder to see the differences between configurations. Importantly, all four configurations are evaluated on the same questions. Config 1's ECE is high because these are difficult questions, but C2, C3, and C4 face the same difficulty. The relative improvements are meaningful within this regime. That said, we agree the scoping has not been stated clearly enough. We will revise the abstract, Section 3.5, and the conclusion to explicitly state that results are specific to the high-disagreement regime and do not characterize performance on the full benchmark distribution. Evaluation on the full distribution is something we will add as a prioritized future work item.
>
> On the single-specialist baseline, we will add a table with per-specialist performance (accuracy, ECE, AUROC) for all four specialists from existing Config 3 data, and add the average single-specialist row to Table 1. We will report these in the revised paper. On compute fairness, Reviewer hgMc is right that Config 4 uses 16x the LLM calls of Config 1. We already discuss the computational overhead as a limitation in Section 5.5, but we acknowledge that a self-consistency baseline with the same compute budget is a fair controlled comparison that we did not run. We will add this as an explicit open empirical question in the revised paper.
>
> For statistical robustness, bootstrap confidence intervals (95%, n=10,000 resamples) are already computed for all metrics and will be added as a table in the revised paper. On MedQA-250 and MedMCQA-250, Config 4 ECE intervals lie entirely below Config 1. On MedMCQA-100 they partially overlap, which we will acknowledge explicitly. On the ECE bin size, the 5-bin design was deliberate. At n=100, finer binning produces empty bins which makes ECE noisier, not more reliable. We will add a note explaining this design choice in the revised paper. We will also foreground the 250-question results as the primary evidence.
>
> For specialty selection and domain mismatch, the four specialties were chosen because they were most frequent in the high-disagreement subsets after the curation pass, this rationale is already stated in Section 3.2 of the submitted paper. On domain mismatch, a gastroenterologist answering a psychiatry question may produce reasoning that lacks domain-specific grounding, leading to higher inconsistency scores not because the answer is wrong but because the agent is reasoning outside its area of expertise. This is a real limitation we will add explicitly to Section 5.5. We are addressing this in ongoing work through an attention-based fusion mechanism that learns to weight specialist contributions based on question content, which would naturally down-weight specialists whose domain is mismatched to the question.